# The Interplay between Mitochondrial Dysfunction and Ferroptosis during Ischemia-Associated Central Nervous System Diseases

**DOI:** 10.3390/brainsci13101367

**Published:** 2023-09-25

**Authors:** He-Yan Tian, Bo-Yang Huang, Hui-Fang Nie, Xiang-Yu Chen, Yue Zhou, Tong Yang, Shao-Wu Cheng, Zhi-Gang Mei, Jin-Wen Ge

**Affiliations:** 1School of Medical Technology and Nursing, Shenzhen Polytechnic University, Xili Lake, Nanshan District, Shenzhen 518000, China; tian903709529@163.com; 2Key Laboratory of Hunan Province for Integrated Traditional Chinese and Western Medicine on Prevention and Treatment of Cardio-Cerebral Diseases, Hunan University of Chinese Medicine, Changsha 410208, China; 3Hunan Academy of Traditional Chinese Medicine, Changsha 410208, China

**Keywords:** cerebral ischemia, central nervous system diseases, mitochondrial dysfunction, ferroptosis, therapeutics

## Abstract

Cerebral ischemia, a leading cause of disability and mortality worldwide, triggers a cascade of molecular and cellular pathologies linked to several central nervous system (CNS) disorders. These disorders primarily encompass ischemic stroke, Alzheimer’s disease (AD), Parkinson’s disease (PD), epilepsy, and other CNS conditions. Despite substantial progress in understanding and treating the underlying pathological processes in various neurological diseases, there is still a notable absence of effective therapeutic approaches aimed specifically at mitigating the damage caused by these illnesses. Remarkably, ischemia causes severe damage to cells in ischemia-associated CNS diseases. Cerebral ischemia initiates oxygen and glucose deprivation, which subsequently promotes mitochondrial dysfunction, including mitochondrial permeability transition pore (MPTP) opening, mitophagy dysfunction, and excessive mitochondrial fission, triggering various forms of cell death such as autophagy, apoptosis, as well as ferroptosis. Ferroptosis, a novel type of regulated cell death (RCD), is characterized by iron-dependent accumulation of lethal reactive oxygen species (ROS) and lipid peroxidation. Mitochondrial dysfunction and ferroptosis both play critical roles in the pathogenic progression of ischemia-associated CNS diseases. In recent years, growing evidence has indicated that mitochondrial dysfunction interplays with ferroptosis to aggravate cerebral ischemia injury. However, the potential connections between mitochondrial dysfunction and ferroptosis in cerebral ischemia have not yet been clarified. Thus, we analyzed the underlying mechanism between mitochondrial dysfunction and ferroptosis in ischemia-associated CNS diseases. We also discovered that GSH depletion and GPX4 inactivation cause lipoxygenase activation and calcium influx following cerebral ischemia injury, resulting in MPTP opening and mitochondrial dysfunction. Additionally, dysfunction in mitochondrial electron transport and an imbalanced fusion-to-fission ratio can lead to the accumulation of ROS and iron overload, which further contribute to the occurrence of ferroptosis. This creates a vicious cycle that continuously worsens cerebral ischemia injury. In this study, our focus is on exploring the interplay between mitochondrial dysfunction and ferroptosis, which may offer new insights into potential therapeutic approaches for the treatment of ischemia-associated CNS diseases.

## 1. Introduction

Cerebral ischemia, characterized by insufficient blood flow and oxygen delivery to the brain, is a major cause of morbidity and mortality worldwide [1]. It can trigger various pathological processes, including inflammation, oxidative stress, excitotoxicity, increased blood–brain barrier (BBB) permeability, and mitochondrial dysfunction. These processes are also implicated in neurodegenerative diseases of the CNS such as AD, PD, and others. There is a wealth of evidence suggesting that cerebral ischemia is associated with the development and progression of CNS diseases including ischemic stroke [2], AD [3], PD [4], epilepsy [5], and other [6] related conditions collectively referred to as ischemia-associated CNS diseases [7].

Mitochondria, as regulators of energy generation, play a crucial role in maintaining cellular homeostasis and function. Additionally, they serve as the primary energy source for the brain, and their dysfunction is closely associated with cerebral ischemic injury, which is essential for the pathophysiology of ischemia-associated CNS diseases [8]. During ischemia, there are transient but significant reductions in adenosine triphosphate (ATP) levels. Mitochondria are pivotal in various cellular functions, including ATP production, maintaining Ca^2+^ homeostasis, generating ROS, and regulating apoptosis. Research has shown that resveratrol can effectively enhance SOD activity and preserve mitochondrial integrity. It achieves this by preventing the loss of mitochondrial membrane potential, ultimately reducing neuronal apoptosis following cerebral ischemia [9]. Furthermore, selenium has been shown to protect against cell death induced by glutamate and hypoxia by reducing glutamate-induced ROS production, preserving mitochondrial membrane potential, and increasing mitochondrial biogenesis [10].

Due to mitochondrial dysfunction and other pathophysiological responses induced by cerebral ischemia, neurons are unable to maintain the normal ionic gradient across the neuronal membrane. This pathological process can lead to various types of cell death, including apoptosis, necrosis, autophagy, and ferroptosis (Figure 1) [11,12,13,14,15]. Ferroptosis is a recently discovered form of regulated cell death characterized by the abnormal accumulation of iron-dependent lipid peroxidation and lipophilic ROS in cellular membranes [16,17]. Previous research has shown that the mechanisms of ferroptosis are involved in glutathione (GSH) depletion [16,18,19], glutathione peroxidase 4 (GPX4) inactivation [16,19,20,21], and metabolic imbalances of iron and lipids [17,19,21]. Previous studies have shown that ferroptosis has pathological roles in a wide variety of diseases, including stroke, PD, and AD [22]. Additionally, activation of ferroptosis has been found to trigger mitochondrial swelling and mitochondrial membrane potential collapse via MPTP opening, ultimately leading to cell death [23]. Increasing evidence has prompted various interplay between ferroptosis and mitochondrial dysfunction following cerebral ischemia. In this case, mitochondrial iron metabolism is impaired, which results in excessive free iron accumulation in mitochondria and lipid peroxidation of their membranes. Another pathway of lipid peroxidation accumulation in the mitochondria is cysteine deprivation, which promotes glutaminolysis, thereby effectively enhancing mitochondrial respiration. Stimulating the activity of the tricarboxylic acid cycle gives rise to mitochondrial hyperpolarization and increased ROS production, which promotes the induction of lipid peroxidation and ferroptosis [24]. The activation of ferroptosis in ischemia-associated CNS diseases creates a vicious cycle of worsening cerebral ischemia injury. Oxidative stress, iron dysregulation, mitochondrial dysfunction, and inflammatory responses feed into one another, amplifying the overall damage to brain tissue. Specific feedback mechanisms such as the depletion of antioxidants and disruption of mitophagy further enhance this detrimental cycle. It is crucial to understand the interplay between mitochondrial dysfunction and ferroptosis in the context of ischemia-associated CNS diseases. In this review, we aim to summarize and discuss the mechanisms underlying this interplay and its relevance to ischemia-associated CNS diseases. By clarifying this mechanism, we hope to provide a new perspective that can contribute to the development of treatments for ischemia-associated CNS diseases.

## 2. Mitochondrial Dysfunction and Ischemia-Associated CNS Diseases

Mitochondria play a critical role not only in other cells but also in neurons, and their structural and functional integrity is particularly important. Cerebral ischemia leads to damage to the structure and function of mitochondria, which is characterized by the opening of MPTP, oxidative stress, and disturbances in mitochondrial quality control. In this context, we aim to investigate these characteristics and their relationships with ischemia-associated CNS diseases.

### 2.1. Mitochondrial Membrane Permeability Transition Pore

Recent evidence suggests that the MPTP is associated with a multi-protein complex formed by ATP synthase. The MPTP complex, which is a non-specific and voltage-dependent channel, consists of the voltage-dependent anion channel (VDAC), adenine nucleotide translocator (ANT), and cyclophilin D (CypD) [25]. The opening of MPTP results in the uncoupling of oxidative phosphorylation, mitochondrial depolarization, swelling, and mechanistic disruption of the outer mitochondrial membrane [26,27,28]. The swollen organelle matrix preceding the uncoupling of oxidative phosphorylation releases cytochrome c, which in turn activates apoptosis-inducing factor (AIF) and endonuclease G (Endo G), subsequently translocating to the nucleus and causing chromatin condensation and deoxyribonucleic acid (DNA) breakage [29].

Previous studies have demonstrated that MPTP opening occurs in response to mitochondrial Ca^2+^ overload [30]. In addition, it also can sense oxidative stress, inflammation, pH imbalance, and iron disturbance generated by ischemic tissue and participate in this process to promote mitochondrial damage in neuronal cells [31,32]. MPTP induction induces the activation of an inflammatory response by the release of succinate and mitochondrial damage-associated molecular patterns (DAMPs) that will induce tissue damage after cerebral ischemia [33]. BCL2 family proteins (Bax, Bid, Puma, and BNIP3) can increase the permeability of the outer mitochondrial membrane, which results in the creation of permeability pores and upregulates other apoptotic factors [27,34]. Moreover, the formation of MPTP is aided by the presence of oxidizing agents, inorganic phosphate (Pi), and alkaline pH [30,35]. Therefore, blocking MPTP to maintain mitochondrial integrity and function may be beneficial in cerebral ischemia.

### 2.2. ROS Production and Mitochondrial Dysfunction

Mitochondria represent a crucial source of ROS, as some of the electrons are instead leaked out of the mitochondrial electron transport chain [36,37]. Under physiological conditions, ROS, as the second messenger involved in a variety of cellular pathways, allows cells to undergo normal physiological activities, and sufficient antioxidant systems limit the damage to cellular components [38]. However, following cerebral ischemia, mitochondrial dysfunction disrupts the balance between ROS production and scavenging, leading to oxidative stress and cell injury [39].

ROS can lead to protein modification or degradation via interacting with amino acids in protein molecules. It can also react with the side chains and the backbone of the protein, causing protein oxidation, peptide bond breaking, and protein degradation [40]. Oxidation of proteins by ROS can result in crosslink and aggregation between proteins, causing their functional alterations, including enzyme inactivation or modification of ion channel activity [41,42,43,44]. Cerebral ischemia leads to extensive protein oxidation in human and animal models of stroke [45,46]. Glutamine synthetase can catalyze the conversion of glutamate to glutamine in astrocytes to protect neurons from excitotoxicity, and oxidative inactivation of glutamine synthetase has been found as a significant factor in the neurotoxicity induced by cerebral ischemia in gerbils [47,48]. ROS surges also disrupt DNA double strands and lead to interchain crosslinks, protein–DNA crosslinks, DNA mutations, and DNA structural alterations, which cause irreversible damage to cells [49]. 8 Hydroxy 2V deoxyguanosine (8OHdG) is a common product of DNA oxidative damage, and elevated levels of 8OHdG during cerebral ischemia highlighted that extensive DNA oxidation in cerebral ischemia precedes DNA fragmentation [46,50,51,52]. In addition, ROS was found to directly damage the mitochondrial proteins and mtDNA, resulting in membrane integrity destruction, which in turn depolarizes and induces cell death [53]. Oxidative degradation of lipids, known as lipid peroxidation, closely involves ferroptosis [54]. ROS can attack lipids containing carbon–carbon double bonds, including polyunsaturated fatty acids (PUFAs) and produce unstable lipid radicals, reacting with oxygen to generate lipid peroxyl radicals. Lipid peroxyl radicals can react with other lipid acids to form another lipid radical and lipid peroxide [40]. The end products of lipid peroxidation-reactive aldehydes are generated in this process, including malondialdehyde (MDA) and 4-hydroxynonenal (4-HNE), which have been employed as markers for lipid peroxidation [55,56,57]. MDA can react with amino acids in proteins and other molecules to form adducts, which can cause protein modification and DNA damage or mutation by promoting intramolecular or intermolecular protein or DNA cross-linking, resulting in secondary injury [58,59,60,61,62]. 4-HNE, a second messenger, can regulate a variety of transcription factors and major cell signaling pathways. Low levels of 4HNE promote cell proliferation and differentiation as well as protect them against free radicals and inflammation. On the other hand, high concentrations of 4HNE cause cell apoptosis. Elevated lipid peroxidation has been found in rat cerebral ischemia models [63,64] and is identified as playing a material role in cell death in cerebral ischemia.

### 2.3. Disturbed Mitochondrial Quality Control

Mitochondrial quality control, including mitochondrial fission and fusion, mitochondrial biogenesis, and mitophagy, is the cornerstone of preserving the integrity and stability of morphology, quantity, and function and plays an indispensable role in alleviating the pathological process of ischemia-associated CNS diseases [65]. However, recent reports found that prolonged ischemic times, directly related to mitochondrial quality control, lead to the disruption of dynamic balance, which ultimately intensifies mitochondrial damage and causes cell death [66].

In order for pre-existing mitochondria to proliferate and divide, a process known as mitochondrial biogenesis coordinates the transcription of mitochondrial and nuclear genes [67]. The peroxisome proliferator-activated receptor-coactivator-1α(PGC-1α) protein family plays an indispensable role in a multistep process that permits coordination [68]. PGC-1α, the primary regulator of mitochondrial biogenesis, is activated during global cerebral ischemia. It stimulates the expression and activation of several nuclear proteins, including UCP2 and SOD2, leading to increased mitochondrial biogenesis and conferring neuroprotective effects [69,70].

Aside from mitochondrial biogenesis, mitochondrial dynamics, which includes morphological changes associated with fission and fusion, is also a precisely regulated mechanism [71]. ATP supply deficiency following cerebral ischemia leads to neuronal plasma membrane depolarization, which leads to glutamate release and subsequent excitotoxicity [72]. Glutamate excitotoxicity influences mitochondrial fission and fusion as well as the imbalance in mitochondrial fission and fusion, resulting in NMDA receptor upregulation and oxidative stress. Additionally, due to increased glutamate release from neurons and astrocytes induced by cerebral ischemia, NMDA and amino-3-hydroxy-5-methyl-4-isoxazole-propionic acid (AMPA) receptors, which are closely associated with ischemia-associated CNS diseases, are overactivated [73,74,75,76,77]. Furthermore, mitochondrial fission and ROS interact with one another. On the one hand, increased mitochondrial fission can enhance ROS; correspondingly, inhibition of mitochondrial fission will restore ROS levels to normal [78]. On the other hand, the increase of ROS can encourage the activation of Drp1, causing an increase in mitochondrial fission, which further increases ROS production, while knocking out Drp1 can reduce mitochondrial fragmentation caused by oxidative stress [79].

Selective mitochondrial autophagy, known as mitophagy, promotes the control of oxidative stress and recovery of damaged mitochondria composition. Excessive mitophagy may provoke uncontrolled degradation of mitochondria, and defective mitophagy may result in the accumulation of mitochondria and excessive ROS production, which has been proposed to contribute to cell degeneration and activation of cell death pathways [80,81]. Moderate autophagy is protective, while excessive autophagy may contribute to cell deaths during ischemia [82]. It has been reported that BNIP3L rescues mitophagy deficiency and protects against cerebral ischemia [83], and inhibiting peroxynitrite-mediated mitochondrial activation could reduce neuronal damage in ischemic stroke [84]. Additionally, previous studies have demonstrated defective mitophagy in AD and PD, resulting in the aggregation of autophagosomes, abnormal endosomes, and abnormal lysosomes, which contributes to the pathology of AD and PD [85,86].

## 3. Ferroptosis and Ischemia-Associated CNS Diseases

After cerebral ischemia, ferroptosis has been shown to have detrimental effects on the brain by promoting rapid neuronal death and dysfunction [87]. Moreover, inhibiting ferroptosis has been found to significantly reduce disease severity and promote functional recovery [88]. Numerous lines of evidence have demonstrated that the mechanisms of ferroptosis involve lipid peroxidation triggered by GSH depletion, inactivation of GPX4, and iron overload, all of which contribute to the development of ischemia-associated CNS diseases [17,89,90]. Cerebral ischemia triggers oxidative stress and ROS production, depleting GSH and inactivating GPX4. This depletion of GSH and GPX4 inhibition promotes lipid peroxidation. Lipoxygenase activation catalyzes lipid peroxidation, while calcium influx exacerbates oxidative stress and lipid peroxidation, creating a vicious cycle that can lead to cell damage and death in ischemic conditions. In this context, we aim to elucidate the mechanisms of ferroptosis and their underlying roles in ischemia-associated CNS diseases.

### 3.1. Lipid Peroxidation

Lipid peroxidation, the prototype of a free radical chain reaction that inserts oxygen into a C–H bond in oxidizable free PUFAs, accumulates and causes ferroptosis [91]. It is noteworthy that among the PUFAs-related phospholipids, phosphatidylethanolamine (PEs) containing arachidonoyl (AA) and its derivative adrenaline adrenoyl (AdA) moieties have been indicated to be the indispensable substrates of oxidation in ferroptosis [90]. An increasing body of work implicates lipid peroxides as important mediators of many pathological states, including stroke, PD, and AD [92].

Ferroptosis induced by lipid peroxidation involves the following steps: acyl-CoA synthetase long-chain family member 4 (ACSL4), acts as a crucial regulator, and catalyzes the esterification of AA or AdA into PEs to form AA-CoA or AdA-CoA. Then, AA-CoA or AdA-CoA can be incorporated into membrane phospholipids with the catalysis of lysophosphatidylcholine acyltransferase 3 (LPCAT3) to generate AA-PE or AdA-PE. Finally, AA-PE and AdA-PE are the favored substrates for oxidation, undergoing transformation by 15-LOX into ferroptosis signals such as PE-AdA-OH and PE-AA-OH, thus actively participating in the execution of ferroptosis [91].

The brain rich in PUFAs and iron is particularly vulnerable to lipid peroxidation due to its low antioxidant defense capacities [93]. Arachidonate 15-lipoxygenase (ALOX15) is an enzyme responsible for oxygenating PUFAs and bio-membranes. It exhibits selective and specific activity in inducing the oxidation of AA/AdA-PE, leading to the enzymatic production of AA/AdA-PE-OOHs [94,95]. PUFAs are the major source of lipid peroxidation in ferroptosis, and ALOX15 can directly oxidize lipid membranes containing PUFAs [96]. ALOX15 was strongly expressed in neurons and endothelial cells following localized ischemia, while its inhibition or deletion prevented the development of edema and preserved the BBB [97,98]. Lipid peroxidation products serve as underlying biomarkers for ischemia-associated CNS diseases. Moreover, the accumulation of lipid peroxidation on the mitochondrial membrane plays a vital role in ferroptosis [99]. One clinical area where lipid peroxidation is especially significant is degenerative disease of the brain and ischemia-associated CNS diseases. The brain consumes a large volume of oxygen and triggers a high quantity of ROS as a via product of ATP synthesis. Membrane phospholipids in the CNS are highly enriched in PUFAs, incorporating them rapidly from free fatty acids [100].

### 3.2. GSH Depletion and GPX4 Inactivation

Ferroptosis is regulated by lipid peroxidation, which may be reversed by GSH production and GPX4 activation. GSH, a tripeptide consisting of glutamate, glycine, and cysteine with sulfhydryl groups, exerts an indispensable duty in free radical scavenging and detoxification through glutathione S-transferase and GPX4 [101]. As a glutamate/cystine antiporter that produces GSH and GPX4, system xc^–^ plays a beneficial role in preventing ferroptosis [102]. However, extracellular glutamate levels will increase during a stroke, causing glutamate toxicity and system xc^–^ impairment. As such, cystine–glutamate exchange will be blocked and the GSH production, and GPX4 activation will be inhibited, which finally initiates ferroptosis [103,104]. Furthermore, in both cerebral patients and MCAO animal models, there is a decrease in the levels of GSH, accompanied by an elevation in lipid peroxidation [105,106]. Depletion of GSH leads to increased levels of ROS, triggering cell ferroptosis and contributing to the degradation of dopaminergic neurons in PD [107]. Temporarily, GSH supplementation or activation of Nrf2 pathway is expected to be an operative therapeutic process for neuroprotection in PD. Furthermore, in ischemic brain tissue of IS model rats, GSH levels were notably diminished. However, when cysteine supplementation was used to replenish depleted GSH levels, or ferroptosis inhibitors were administered, there was a significant reduction in neuronal injuries following IS [108].

GPX4, a type of lipid enzyme, converts lipid hydroperoxides into non-toxic lipid alcohols, thereby preventing the accumulation of harmful lipid oxidation products [109,110]. Deletion or inactivation of GPX4 can lead to lethal ferroptosis and neurological dysfunction [111,112], often manifesting as progressive cognitive impairment and impaired behavior in the context of cerebral ischemia [113,114]. It has been shown that the expression level of GPX4 was significantly reduced during the acute phase of ischemic stroke and that increasing GPX4 levels can protect neurons from ferroptosis [115]. Additionally, supplementing with compounds such as tetrahydroxy stilbene glycoside and γ-glutamylcysteine can help restore the GSH-GPX4 antioxidant system in AD patients. This restoration leads to a reduction in ROS levels, ultimately mitigating brain injury induced by Aβ in AD patients [116]. Meanwhile, GPX4 expression is decreased, and lipid peroxidation appears in the hippocampus of epileptic model rats, which could be reversed by Fer-1 [117].

### 3.3. Iron Overload

There are two main sources of iron: food-derived iron and iron generated by the hemoglobin of senescent erythrocytes [118]. The Fe^2+^ is absorbed by the small intestinal mucosa epithelial cells, and then, the absorbed Fe^2+^ is oxidized to Fe^3+^ by the ferroxidase hephaestin, and eventually, Fe^3+^ enters the circulation by ferroportin (FPN) [119,120,121]. Furthermore, virtually all circulating Fe^3+^ tightly combines with transferrin (TF) to form the complete TF, which can bind to the membrane protein transferrin receptor 1 (TFR1) to transport the Fe^3+^ into the acidified endosomes [122]. Finally, iron reductase reduces Fe^3+^ to Fe^2+^, which is transported into the labile iron pool in the cytoplasm by divalent metal transporter 1 (DMT1) [123]. Most released Fe^2+^ is utilized in several physiological processes, including the Fenton reaction, DNA synthesis, and mitochondrial oxidative metabolism. Ferritin, a protein with two subunits, namely ferritin heavy chain 1 (FTH1) and ferritin light chain (FTL), can store part of the released Fe^2+^. At the same time, FPN allows other iron that is not utilized or stored to be exported to the extracellular environment. Subsequently, the ceruloplasmin (Cp) can oxidize the exported Fe^2+^, which is bound to serum TF again [124]. Under physiological conditions, iron homeostasis is kept through these intricate networks.

Prior to the concept of ferroptosis, it was already acknowledged that the accumulation of iron in the brain was one of the causes of secondary brain damage [125,126]. Physiologically, BBB can rescue the brain against undulations in systemic iron [127]. However, the BBB is disrupted in acute ischemia, allowing free iron and ferritin to enter the brain parenchyma, and hydrogen peroxide is converted to hydroxyl radicals through the Fenton reaction [128,129]. Furthermore, the acidic environment of cerebral ischemia triggers the dissociation of iron from TF, leading to elevated extracellular iron levels and its transfer to neurons, thereby increasing intracellular iron uptake [130,131,132]. Ferritin, TFR1, and DMT1 have all been shown to increase during cerebral ischemia, potentially contributing to iron uptake [108,133]. Meanwhile, FPN, the only protein that exports iron from cells, is downregulated following cerebral ischemia, further aggravating intracellular iron overload by preventing its outflow [134].

Iron overload has been identified as a significant provenience of oxidative stress in cerebral ischemia. Iron deposits heavily in the basal ganglia, thalami, periventricular, and subcortical white matter regions following severe ischemic and hypoxic brain injury [135]. This process significantly elevates ROS production, leading to direct damage to proteins, amino acids, nucleic acids, and membrane lipids, ultimately mediating ferroptosis. Furthermore, lipid peroxidation originates from three pathways: lipid autoxidation catalyzed by iron, the esterification and oxygenation of PUFAs, and the generation of lipid ROS associated with the Fenton reaction [136,137]. Iron participates in all three pathways, which are critical factors in triggering ferroptosis. It has been shown that multiple iron chelates or iron export compounds have indicated neuroprotective effects in ischemic strokes models [138], supporting iron accumulation as a therapeutic target in ischemic strokes.

## 4. The Interplay between Mitochondrial Dysfunction and Ferroptosis in Ischemia-Associated CNS Diseases

Mitochondria play a crucial role in cellular stress response and are dynamically involved in multiple types of regulated cell death mechanisms, including ferroptosis. Furthermore, there is mounting evidence from various sources, including structural, functional, and autophagic studies, indicating the existence of crosstalk between mitochondria and ferroptosis. This crosstalk is illustrated in Figure 2.

### 4.1. The Interplay between Mitochondrial Dysfunction and Ferroptosis in IS

Based on existing animal studies, mitochondrial dysfunction has been reported in ischemic stroke, and this dysfunction has been correlated with oxygen and glucose deprivation (OGD) [139]. Although a vast array of evidence has suggested that the Fenton reaction is the primary source of ROS in ferroptosis [140,141], mitochondrial ROS production likely contributes to ferroptosis induction by promoting lipid peroxidation [142]. In the ischemic brain, p53 rapidly accumulates and activates transcriptional-dependent and transcriptional-independent programs to cause neuronal apoptosis [143]. Recent studies revealed that p53 favors mitochondrial respiration and contributes to ROS-mediated ferroptosis, which has a direct impact on metabolic versatility [144]. Moreover, MitoROS may activate ER stress, which triggers autophagy and ferroptosis [145]. ONOO-mediated mitophagy is suspected to be excessive in stroke and could provoke uncontrolled degradation of mitochondria, resulting in stored iron releasing, which triggers iron overload and ferroptosis [146,147]. Additionally, the NOD LRR pyrin domain-containing protein 3 (NLRP3) inflammasome in microglia is activated immediately following cerebral I/R injury, and mitochondrial dysfunction plays a key role via releasing mitochondrial ROS, mitochondrial DNA damage, and releasing phospholipid cardiolipin in this process [148,149]. Exceptional activation of the NLRP3 inflammasome via mitochondrial dysfunction is conducive to not merely pyroptosis but also other kinds of cell death, including ferroptosis [150].

The characteristic morphological changes of mitochondria in ferroptosis are the shrunken mitochondria, ruptured external membrane, reduced crista, a compressed internal membrane, and an electron lucent nucleus [151]. These changes eventually lead to cell membrane rupture, mitochondrial membrane potential depolarization, and MPTP opening. MPTP is deeply involved in ROS production in the mitochondria of neuronal cells during ischemic stroke [32]. Moreover, MPTP opening causes mitochondrial energetic dysfunction, organelle swelling and rupture, as well as ferroptosis typically, thus creating a vicious circle [23,152]. These studies suggest that MPTP is an important intersection point between mitochondrial dysfunction and ferroptosis. VDACs, a channel-forming protein situated at the mitochondrial outer membrane, allow the metabolites and irons across the outer membrane [153,154]. The partial closure of VDACs after global ischemia inhibits mitochondrial metabolism and reduces mitochondrial membrane potential [155,156]. It is intriguing that the ferroptosis activator erastin induces the opening of VDAC2/3, facilitating the uptake of mitochondrial iron and elevating the mitochondrial membrane potential. Subsequently, in the following hours, mitochondrial depolarization is observed, signifying a manifestation of mitochondrial dysfunction [24,157,158]. Therefore, stabilizing VADCs can be used as an effective target in ferroptosis treatment.

Additionally, the mitochondrial-dependent proapoptotic protein Bid is transactivated after IR, which may be caused by lipid peroxidation and the increase of ROS induced by ferroptosis. As a result, the mitochondrial membrane potential and integrity are lost, ATP synthesis is impaired, ROS production increases, and AIF is released from the mitochondria to the nucleus [159,160].

### 4.2. The Interplay between Mitochondrial Dysfunction and Ferroptosis in AD

In fact, it is widely acknowledged that mitochondria play a role in AD. Mitochondrial ferritin (FtMt) regulates iron metabolism by controlling the transfer of iron from mitochondria to the cytoplasm, thereby safeguarding mitochondria against oxidative damage induced by excessive iron [161]. And lack of FtMt aggravates amyloid β-induced neurotoxicity. Therefore, we speculate that lack of FtMt may lead to iron overload by aggravating mitochondrial dysfunction via enhancing oxidative stress in AD [162,163]. Iron regulatory protein 1 (IRP1) activation is secondary to mitochondrial dysfunction and is regulated by the mitochondrial ISC biogenesis pathway; it selectively regulates amyloid β precursor protein (APP) mRNA and causes iron accumulation [164,165,166]. To prevent excess cytoplasmic iron, mitoferrin1/2 allows iron to enter the mitochondria to maintain iron homeostasis. However, recent studies have shown that mitoferrin1/2 is unregulated in AD, causing cells to exhibit iron overload [151,167]. Additionally, the pathophysiology of AD has also been linked to the aberrant activation of the NLRP3 inflammasome induced by mitochondrial dysfunction. The NLRP3 inflammasome has been shown to be critical for the pathological drive of amyloid and tau proteins [168]. Iron overload is the most apparent alteration of these pathological processes, which can promote ferroptosis by causing lipid peroxidation via intracellular ROS generated by the Fenton reaction. Mitophagy can reduce mitochondrial ROS production and degrade dysfunctional mitochondria [169]. Thus far, Pink1-Parkin-mediated mitophagy is the most deeply understood mechanism of mitophagy in mammalian cells [86]. Pink1-Parkin-mediated mitophagy, a specific form of autophagy, has been found to be a dysfunctional mitophagic process at the onset of AD [170,171]. Cytosolic Parkin levels decrease during disease progression, resulting in increased mitochondrial dysfunction. Impaired mitophagy may lead to mitochondrial accumulation, increased oxygen consumption, and excessive formation of ROS, which is associated with lipid peroxidation [172]. Additionally, microwave exposure has occasionally been associated with AD. Microwaves, which affect the CNS, may have adverse effects on brain function, including the modulation of neurotransmitters that facilitate signal transmission within the body [173].

A plethora of evidence has shown that lipid peroxidation and GSH depletion, the primary signatures of ferroptosis, have been described in AD [174]. And iron-induced oxidative stress also is an important pathological mechanism of AD that can directly cause neuronal damage and cell death. The ferroptosis pathway may contribute to our understanding of how iron enhances the neurotoxicity of other major pathological features of AD, including those associated with Aβ and tau [175]. Anna et al. found a significant decrease in the activity of GPX in AD [176]. Consistent with this, adult mice with conditional GPX4 ablation in their forebrain neurons have cognitive deficits and hippocampus degeneration resembling AD. It is worth noting that lipid peroxidation and mitochondrial impairments associated with ferroptosis are observed in this model. These effects can be partially mitigated by ferroptosis inhibitors, providing confirmation of the involvement of ferroptosis [114]. Moreover, under iron overload conditions, mtDNA is damaged by ROS produced by the labile iron pool [177]. MtDNA double-strand breaks as well as progressive loss of intact mtDNA, reduced mtDNA transcription, and decreased expression of respiratory chain subunits encoded by the mitochondrial genome were reported in AD cells, all changes that were also observed in mitochondrial ferroptosis. Therefore, we speculate that one of the main culprits of mtDNA damage in AD cells is ferroptosis [178,179].

### 4.3. The Interplay between Mitochondrial Dysfunction and Ferroptosis in PD

Mitochondrial dysfunction is a key pathophysiological change in PD and is an essential initiator of dopaminergic neuron loss [180]. Excessive ROS levels in PD patients trigger mitophagy, which is a negative feedback mechanism crucial for reducing oxidative stress by inhibiting ROS production. Consequently, mitophagy is considered neuroprotective, as it helps eliminate ROS generation. [181,182,183]. However, dysfunction in mitophagy has been observed in several PD models [184], leading to an inability to effectively scavenge ROS. Excessive ROS expedites oxidative stress to cell components and causes extensive lipid peroxidation, leading to ferroptosis [185]. MitoNEET, an iron-containing outer mitochondrial membrane protein, participates in exporting iron from mitochondria [186]. MitoNEET KO mice exhibit multiple characteristics of early neurodegeneration in PD, including mitochondrial dysfunction and loss of striatal dopamine and tyrosine hydroxylase [187]. Knockout of MitoNEET increases the mitochondrial iron content and lipid peroxidation [188], aggravating the adverse effects of erastin-induced ferroptosis. NEET proteins were identified as a newly discovered iron–sulfur protein (2Fe-2S) that mediates the export of sulfur ions and iron between the cytosol and the mitochondria when the synthesis of Fe-S clusters is disturbed [186]. Deletion of one of the mitoNEET isoforms, CDGSH iron sulfur domain 1 (CISD1), leads to ferroptosis by causing mitochondrial iron accumulation and the production of mitochondrial lipid peroxides [188]. Interestingly, CISD1 knockout mice exhibit many features of PD. In addition, recent studies also highlight that Mfrn1/2 is impaired not only in AD but also in PD, which are all linked to ferroptosis. These findings illustrate mitochondria play a crucial role in cellular iron homeostasis, and the regulation of iron storage in mitochondria will be an effective strategy to inhibit ferroptosis in PD.

Intracellular calcium overload interacts with iron overload, which is characteristic of ferroptosis, and ion disorderliness occurs in both PD and AD [189]. Increased cytoplasmic Ca^2+^ promotes 12/15-LOX activation [190]. Furthermore, excess Ca^2+^ entering the mitochondria produces mitochondrial dysfunction and damage, whereas iron overload induces oxidative stress, which increases the flow of excessive calcium signals [191]. In addition, in the experimental model of arsenite-induced neuronal cell death that has been shown to be associated with PD and AD, it was shown that the loss of neurons is caused by the generation of ROS, the imbalance of GSH and GSSG, and ferroptosis-related signaling pathways. Of note, in this model, VADCs and mitogen-activated protein kinases were activated, leading to mitochondrial dysfunction, which may be associated with ferroptosis. Despite the clinical distinctions between AD and PD, these disorders share partially similar clinicopathologic features in ferroptosis, including unregulated mitoferrin1/2, calcium overload, iron overload, and VADCs opening.

### 4.4. The Interplay between Mitochondrial Dysfunction and Ferroptosis in Epilepsy

Mitochondrial oxidative stress induced by ROS is emerging as a crucial element involved in the onset of epilepsy [192]. Dysregulation of astrocyte glutamate metabolism due to mitochondrial oxidative stress results in high extracellular levels of glutamate occurring in the brain during epileptic seizures, which can cause recurrent seizures [193,194,195]. Moreover, excessive glutamate induces neuronal over-excitation and neuronal excitotoxicity and triggers ferroptosis [17]. Additionally, previous studies have discovered that autophagy induced by mitochondrial oxidative stress is crosstalk with ferroptosis. Mitochondrial antioxidants such as mito-TEMPO can reduce GSH depletion and autophagic response, which induces autophagy inhibition to block the ferroptosis process [196].

GSH serves as an imperative free radical scavenger compound, which plays an imperative role in alleviating oxidative stress and preventing neuronal death [197]. It was found to reverse mitochondrial dysfunction [198]. Patients with epilepsy exhibit lower levels of GPX4 and GSH compared to normal controls, and these alterations have been associated with ferroptosis [199]. The persistently low GPX4 and GSH result in neuronal excitability changes, hippocampal neuron loss, and astrocyte proliferation, which may induce mitochondrial dysfunction [200]. Meanwhile, in rats with temporal lobe epilepsy (TLE) induced by kainic acid, a considerable accumulation of lipid peroxidation and consumption of GSH have been observed, combined with a decrease in the mitochondrial region of hippocampal neurons [201]. Collectively, these findings imply that mitochondrial dysfunction due to low GPX4 and GSH levels during ferroptosis is a nonnegligible reason for cell damage in epilepsy. Particularly, the p53 pathway, which can adjust necrosis and autophagic activity, including mitophagy, directly alters the metabolic versatility by promoting mitochondrial respiration and contributing to the ROS-mediated ferroptosis in ischemia-associated CNS diseases [144,202,203,204,205].

## 5. Targeting Intervention of Mitochondrial Dysfunction and Ferroptosis for the Treatment against Ischemia-Associated CNS Diseases

Recent studies have revealed that mitochondrial dysfunction and ferroptosis are not only associated with mechanisms of neuronal death but also have significant clinical applications in the diagnosis and treatment of ischemia-associated CNS diseases. As a result, numerous studies have focused on this aspect and have suggested that the potential mechanism for preventing and treating ischemia-associated CNS diseases may involve the intervention of mitochondrial dysfunction and ferroptosis. This relationship is illustrated in Table 1.

Mitochondria are double-membrane organelles, featuring functionally and structurally distinct outer and inner membranes that create a barrier between the intermembrane space and the matrix. Persistent mitochondrial damage disrupts energy metabolism, resulting in decreased ATP production, elevated ROS levels, and impaired calcium buffering. These factors collectively contribute to the neuronal loss observed in ischemia-associated CNS diseases. It has been reported that α-lipoic acid (LA) supplementation effectively inhibited the high phosphorylation of Tau in several AD-related sites and reversed the cognitive decline of P301S Tau transgenic mice. When studying the death patterns of various neural cells in the brain tissues of mice treated with LA, it was found that Tau-induced iron overload, lipid peroxidation, and inflammation related to ferroptosis were significantly blocked by LA administration, suggesting that LA plays a role in inhibiting Tau hyperphosphorylation and ferroptosis [208]. Moreover, LA, as a “mitochondrial nutraceutical”, cannot only improve the antioxidant system and stimulate mitochondrial biogenesis but also directly prevent ROS production using the thiol group for redox regulation. LA also can be inducing the overexpression of Nrf2, which promotes improved mitochondrial function in neurons [209]. Idebenone, an analog of the well-known antioxidant compound coenzyme Q10 (CoQ10), rescues cells against ferroptosis in PD by decreasing the striatal levels of the lipid peroxidation products and increasing the expression of GPX4 [210]. Meanwhile, idebenone can modulate the expression of outer membrane proteins related to mitophagy, such as VDAC1 and BNIP3. It activates the Parkin/PINK1 mitophagy pathway, facilitating the removal of damaged mitochondria and thereby reducing dopaminergic neuron damage. This improvement in neuronal health leads to a reduction in behavioral disorders observed in mice with PD [211]. Clinical phase therapeutic vatiquinone (EPI-743, α-tocotrienol quinone) has decreased the frequency and associated morbidity of mitochondrial disease. Further research indicates that EPI-743 reduces the incidence of epilepsy by protecting from ferroptosis under GSH depletion and iron overload conditions [213]. In addition, RTA 408 activates Nrf2 via blocking Keap1, which prevents mitochondrial depolarization, ferroptosis, and lipid peroxidation in an in vitro model of seizure-like activity [212].

BID-mediated mitochondrial damage is a prerequisite for death signaling in neurons in severe oxidative stress paradigms. Recent studies have found that erastin-induced ferroptosis in neuronal cells was accompanied by BID transactivation to mitochondria, loss of mitochondrial membrane potential, enhanced mitochondrial fragmentation, and decreased ATP levels. Bid knockout can prevent mitochondrial dysfunction in ferroptosis paradigms. Conversely, the ferroptosis inhibitor ferrostatin-1 blocked BID transactivation, mitochondrial fragmentation, and the loss of mitochondrial membrane potential following glutamate exposure in neuronal cells. This confirms that ferroptosis shares detrimental mechanisms of ROS formation upstream of BID-dependent mitochondrial deterioration [160]. These findings show that mitochondrial transactivation of BID links ferroptosis to mitochondrial dysfunction as the final execution step in this paradigm of oxidative cell death. Nrf2 is involved in modulating ferroptosis-related genes, including genes for GSH regulation (synthesis, cysteine supply via system xc^–^, GSH reductase, and GPX4), NADPH regeneration that is crucial for Gpx4 activity (glucose 6-phosphate dehydrogenase, phosphogluconate dehydrogenase, and malic enzyme), and iron regulation (iron export and reserve, heme synthesis, and catabolism) [214,215,216,217]. Peroxisome proliferator-activated receptor gamma (PPARγ), a key regulator of lipid metabolism [218], can be activated by oxidized lipids that are relevant to the initiation of ferroptosis [219]. This activation is reciprocally regulated with Nrf2 and plays a role in lipid regulation [218,220]. Additionally, Nrf2 also plays an important role in modulating the mitochondrial quality control system, including biogenesis via interaction with PGC-1α and mitophagy via a P62-dependent, PINK1/Parkin-independent mechanism [221,222]. Impaired mitochondrial function was observed in Nrf2 knockout mice, whereas activation of Nrf2 enhanced mitochondrial function and resistance to stressors [223,224,225]. Although BID and Nrf2 have not been thoroughly studied, current evidence suggests that BID and Nrf2 are effective targets for intervention in ferroptosis and mitochondrial dysfunction. Further studies to determine the neuroprotective effects of BID and Nrf2 may be a beacon for future treatment of ischemia-associated CNS diseases.

## 6. Conclusions and Perspectives

In summary, the complex interplay between mitochondrial dysfunction and ferroptosis is a promising avenue of research in the context of ischemia-associated CNS diseases. Ischemic events trigger a series of molecular and cellular processes, including mitochondrial dysfunction and the initiation of ferroptosis, which are closely intertwined and contribute to the observed neuropathological outcomes in conditions such as stroke and ischemic brain injury. Mitochondrial dysfunction during ischemia, characterized by factors like membrane potential loss, heightened oxidative stress, and structural alterations, significantly exacerbates cellular damage. It serves as a critical precursor to ferroptosis, which is a regulated cell death mechanism associated with iron accumulation, lipid peroxidation, and oxidative stress. The accumulation of iron within mitochondria during ischemic events further intensifies oxidative stress, potentially expediting the onset of ferroptosis and compromising neuronal survival. The regulatory mechanisms controlling mitochondrial dynamics (fusion and fission) involve various proteins and signaling pathways. Dysregulation of these processes can contribute to pathologies by impacting energy production, cellular homeostasis, and oxidative stress. The significance of iron metabolism as a crucial link between mitochondrial dysfunction and ferroptotic cell death in CNS diseases is evident.

The interaction between mitochondrial dysfunction and ferroptosis presents exciting prospects for the development of innovative therapeutic approaches. Exploring key components in these processes, such as regulators of iron metabolism, antioxidants, and modulators of mitophagy, may open doors to potential strategies for safeguarding neurological function and promoting recovery in ischemia-associated CNS disorders. However, it is important to acknowledge that numerous unanswered questions persist. Further investigation is essential to fully comprehend the intricacies of these intertwined pathways. Moreover, the translation of these research findings into effective clinical interventions remains a formidable challenge. Nonetheless, the ongoing exploration of the crossroads between mitochondrial dysfunction and ferroptosis offers optimism for advancing our understanding of CNS diseases related to ischemia, potentially leading to improved patient outcomes in the future.

## Figures and Tables

**Figure 1 brainsci-13-01367-f001:**
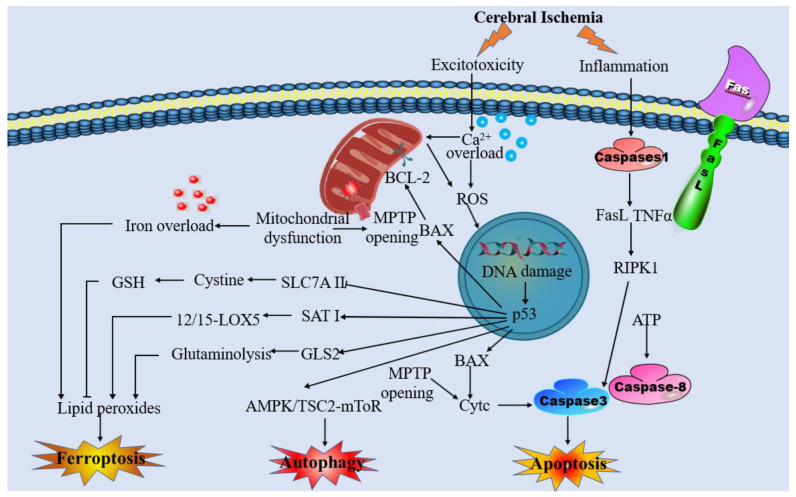
Overview of the pathophysiology of cerebral ischemia.

**Figure 2 brainsci-13-01367-f002:**
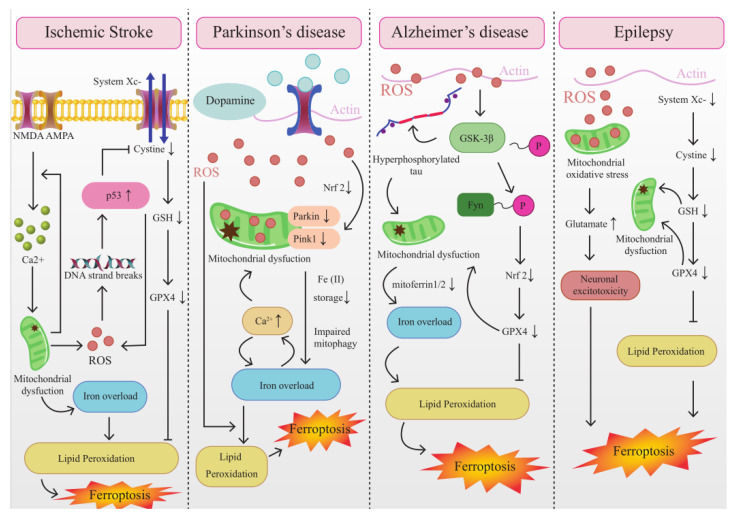
The Interplay between mitochondrial dysfunction and ferroptosis in ischemia-associated CNS diseases.

**Table 1 brainsci-13-01367-t001:** Studies on ischemic encephalopathy treatment regarding mitochondrial dysfunction and ferroptosis.

Interventions	Mechanism for Mitochondrial Dysfunction	Mechanism for Ferroptosis	References
Kaempferol	Inhibit mitochondrial fission Maintain mitochondrial HK-II through activation of Akt	Activate Nrf2, SLC7A11, and GPX4	[206,207]
α-Lipoic acid (LA)	Improve the antioxidant system, stimulating mitochondrial biogenesis Prevent ROS production, and inducing Nrf2 overexpression	Block Tau-induced iron overload, lipid peroxidation, and inflammation	[208,209]
Idebenone	Regulate the expression of the VDAC1 and BNIP3	Decrease lipid peroxidation	[210,211]
RTA 408	Activate NRF2 by inhibiting Keap1 Inhibit mitochondrial depolarization	Activate NRF2 by inhibiting Keap1 Inhibit lipid peroxidation and ferroptosis	[212]

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
