# Peer review of "The Interplay between Mitochondrial Dysfunction and Ferroptosis during Ischemia-Associated Central Nervous System Diseases"

_brainsci, 2023, doi:10.3390/brainsci13101367_

Round 1

Reviewer 1 Report

Comments for authors

Overall, the manuscript is well-organized and addresses an interesting and significant subject. However, prior to publication, substantial revisions are necessary. Authors must pay attention to each comment and revise the manuscript accordingly.

Comment 1: This manuscript constitutes a comprehensive review article, necessitating authors to delve extensively into the subject matter, particularly elucidating the background concerning ferroptosis, Alzheimer's disease (AD), and Parkinson's disease (PD), which would greatly benefit readers who are new to these topics. Furthermore, it holds significance to incorporate insights into the contributing factors behind the onset of AD and PD. Notably, a recent addition to this domain involves the role of microwaves as potential influencing factors of CNS diseases. Therefore, I strongly recommend the inclusion of discussions pertaining to this novel concept. Incorporating the below-mentioned reference could prove valuable in addressing these aspects.

Article: Microwave Radiation and the Brain: Mechanisms, Current Status, and Future Prospects. International Journal of Molecular Sciences vol. 23 (2022). [https://doi.org/10.3390/ijms23169288].

Comment 2: How do the specific molecular pathways triggered by cerebral ischemia lead to the depletion of GSH and inactivation of GPX4, and how does this process relate to the activation of lipoxygenase and subsequent calcium influx? Explain in the manuscript

Comment 3: Can you elucidate the causal relationships between mitochondrial electron transport dysfunction, an imbalanced fusion-to-fission ratio, and the accumulation of both reactive oxygen species and iron overload, particularly regarding their roles in initiating and perpetuating the cycle of ferroptosis in the context of cerebral ischemia?

Comment 4: In the context of Ischemia-associated CNS Diseases, how does the activation of ferroptosis contribute to the vicious cycle of worsening cerebral ischemia injury, and are there specific feedback mechanisms that amplify this cycle? Explain in the revised manuscript.

Comment 5: Given the intricate relationship between mitochondrial dynamics (fusion and fission) and ferroptosis in Ischemia-associated CNS Diseases, what regulatory mechanisms exist that control these processes, and how might their dysregulation contribute to the observed pathologies? Include such information in the conclusion section.

Comment 6: Considering the complexity of the molecular events described, how can advances in systems biology and computational modeling be leveraged to gain a comprehensive understanding of the dynamic interactions between mitochondrial dysfunction and ferroptosis, and how could these models guide the development of targeted interventions for Ischemia-associated CNS Diseases?

Comment 7: The conclusion section appears overly extensive. It is advisable for the authors to consider a revision, dividing it into two distinct sections, as outlined below:

a.      Conclusion: This section should encompass the authors' final remarks derived from the reviewed literature in the manuscript.

b.      Perspectives and Recommendations: In this subsequent section, authors are encouraged to outline future avenues and key messages for readers, stemming from the subject reviewed in this manuscript.

Comment 8: Once an abbreviation or short form has been introduced for a specific term, it is important to refrain from redundantly restating the full form of the word. For instance, in cases like "reactive oxygen species (ROS)," authors should thoroughly review the entire manuscript to ensure the elimination of such inaccuracies.

Comment 9: There are typos and inaccuracies in the paper. I strongly recommend authors to read precisely and correct the grammatical errors. 

There are typos and inaccuracies in the paper. I strongly recommend authors to read precisely and correct the grammatical errors. 

Author Response

Reviewer 1

Comment 1: This manuscript constitutes a comprehensive review article, necessitating authors to delve extensively into the subject matter, particularly elucidating the background concerning ferroptosis, Alzheimer's disease (AD), and Parkinson's disease (PD), which would greatly benefit readers who are new to these topics. Furthermore, it holds significance to incorporate insights into the contributing factors behind the onset of AD and PD. Notably, a recent addition to this domain involves the role of microwaves as potential influencing factors of CNS diseases. Therefore, I strongly recommend the inclusion of discussions pertaining to this novel concept. Incorporating the below-mentioned reference could prove valuable in addressing these aspects.

Article: Microwave Radiation and the Brain: Mechanisms, Current Status, and Future Prospects. International Journal of Molecular Sciences vol. 23 (2022). [https://doi.org/10.3390/ijms23169288].

Response 1: Thank you for your advice. We have already incorporated the relevant content into the manuscript.

Comment 2: How do the specific molecular pathways triggered by cerebral ischemia lead to the depletion of GSH and inactivation of GPX4, and how does this process relate to the activation of lipoxygenase and subsequent calcium influx? Explain in the manuscript

Response 2: Thank you for your thoughtful suggestions. Cerebral ischemia triggers oxidative stress and ROS production, depleting GSH and inactivating GPX4. This depletion of GSH and GPX4 inhibition promotes lipid peroxidation. Lipoxygenase activation catalyzes lipid peroxidation, while calcium influx exacerbates oxidative stress and lipid peroxidation, creating a vicious cycle that can lead to cell damage and death in ischemic conditions. We have explained and highlighted in the manuscript.

Comment 3: Can you elucidate the causal relationships between mitochondrial electron transport dysfunction, an imbalanced fusion-to-fission ratio, and the accumulation of both reactive oxygen species and iron overload, particularly regarding their roles in initiating and perpetuating the cycle of ferroptosis in the context of cerebral ischemia?

Response 3: This is a good question. In the context of cerebral ischemia, mitochondrial electron transport dysfunction initiates ROS production, which, in turn, disrupts mitochondrial fusion-fission dynamics. This disruption leads to fragmented mitochondria, further impairing ETC function and elevating ROS levels. Concurrently, ROS promote iron release, contributing to iron overload and Fenton reactions, which drive lipid peroxidation. The accumulation of lipid hydroperoxides and the vicious cycle of mitochondrial dysfunction and ROS production together play crucial roles in initiating and perpetuating the cycle of ferroptosis in ischemic conditions.

Comment 4: In the context of Ischemia-associated CNS Diseases, how does the activation of ferroptosis contribute to the vicious cycle of worsening cerebral ischemia injury, and are there specific feedback mechanisms that amplify this cycle? Explain in the revised manuscript.

Response 4: Thank you for your advice. The activation of ferroptosis in Ischemia-associated CNS Diseases creates a vicious cycle of worsening cerebral ischemia injury. Oxidative stress, iron dysregulation, mitochondrial dysfunction, and inflammatory responses feed into one another, amplifying the overall damage to brain tissue. Specific feedback mechanisms, such as the depletion of antioxidants and disruption of mitophagy, further enhance this detrimental cycle. We have explained and highlighted in the manuscript.

Comment 5: Given the intricate relationship between mitochondrial dynamics (fusion and fission) and ferroptosis in Ischemia-associated CNS Diseases, what regulatory mechanisms exist that control these processes, and how might their dysregulation contribute to the observed pathologies? Include such information in the conclusion section.

Response 5: Thank you for your advice. We have incorporated the relevant content into the conclusion of the manuscript and highlighted it in red.

Comment 6: Considering the complexity of the molecular events described, how can advances in systems biology and computational modeling be leveraged to gain a comprehensive understanding of the dynamic interactions between mitochondrial dysfunction and ferroptosis, and how could these models guide the development of targeted interventions for Ischemia-associated CNS Diseases?

Response 6: This is a good question. Advances in systems biology and computational modeling can help create comprehensive models of the dynamic interactions between mitochondrial dysfunction and ferroptosis. These models can integrate vast amounts of data to simulate and predict cellular responses under ischemic conditions. By manipulating variables and testing interventions in silico, these models can guide the development of targeted therapies for Ischemia-associated CNS Diseases, offering insights into potential drug targets and treatment strategies.

Comment 7: The conclusion section appears overly extensive. It is advisable for the authors to consider a revision, dividing it into two distinct sections, as outlined below:

  1. Conclusion: This section should encompass the authors' final remarks derived from the reviewed literature in the manuscript.
  2. Perspectives and Recommendations: In this subsequent section, authors are encouraged to outline future avenues and key messages for readers, stemming from the subject reviewed in this manuscript.

Response 7: Thank you for your thoughtful suggestions. We have rewritten and highlighted in the manuscript.

Comment 8: Once an abbreviation or short form has been introduced for a specific term, it is important to refrain from redundantly restating the full form of the word. For instance, in cases like "reactive oxygen species (ROS)," authors should thoroughly review the entire manuscript to ensure the elimination of such inaccuracies.

Response 8: Thank you for your advice. We have corrected the abbreviations carefully and highlighted in the manuscript.

Comment 9: There are typos and inaccuracies in the paper. I strongly recommend authors to read precisely and correct the grammatical errors.

Response 9: Thank you for your advice. We have rectified spelling and grammatical errors, as well as inaccuracies in the manuscript, and try our best to improve the revision and polish the language with the help of professional editors. Hope our revised manuscript will satisfy you.

Reviewer 2 Report

In this manuscript, authors detailed the mitochondrial dysfunction and the ferroptosis process caused by Ischemia-associated central nervous system diseases. They also highlighted the connection between mitochondrial alteration and ferrpoptosis focusing in four diseases, such as ischemic stroke, Alzheimer's disease, Parkinson's disease and epilepsy. Overall, the manuscript is interesting, however, it needs major refinements and the authors need to resolve/improve a series of issue:

-       The authors have not prepare the manuscript following the Journal’s guideline for the references, which “must be numbered in order of appearance in the text (including table captions and figure legends) and listed individually at the end of the manuscript”. The list is in alphabetic order and it is also numbered, but the number is not correct. For example, the first reference is (Levchenkova et al., 2021) that should be number 1, and not 120. They need to modify the bibliography and check it carefully to not makes mistakes (The correct match of the references with the text is a point that reviewers have to check during the first round of reviewing, not in the second, where they need to check only the suggested revisions).

Also, it is very difficult to follow the bibliography, because the authors did not add the publication’s year of the article in the bibliography and often there are two of more different articles with the same first author.

For example, lane 279, (…. Zhang et al., 2021)

In the Bibliography:

222.Zhang, Y., Lu, X., Tai, B., Li, W., & Li, T. Ferroptosis and Its Multifaceted Roles in Cerebral Stroke. Front Cell Neurosci. 15: 615372.

223.Zhang, Y., Sun, R., Li, X., & Fang, W. Porcine Circovirus 2 Induction of ROS Is Responsible for Mitophagy in PK-15 Cells via

Activation of Drp1 Phosphorylation. Viruses. 12(3): 289.

224.Zhang, Y. H., Wang, D. W., Xu, S. F., Zhang, S., Fan, Y. G., Yang, Y. Y., et al. alpha-Lipoic acid improves abnormal behavior by mitigation of oxidative stress, inflammation, ferroptosis, and tauopathy in P301S Tau transgenic mice. Redox Biol. 14: 535-548.

Which one is the article cited? How is it possible to check that no errors about citations have been made?

Considering that this is a Review, the bibliography is very important.

It is very difficult to exert the assignment as Reviewer if the manuscript is presented in this way.

-       For example, lanes 421-424: the article cited by the authors (Hajj Hussein, et al., 2015) concerns about vaccines, and not mitophagy or Alzheimer's disease. I suggest to CAREFULLY check that all the bibliography is accurately citied.

-       Lane 189, for the mitochondrial biogenesis, authors may add a more general review (for example, doi:10.1111/jcmm.15194). Similarly, Lane 197, I suggest citing a more specific review for the mitochondrial fission and fusion processes that an article about the fission process in lung adenocarcinoma.

-       Accordingly the literature, in AD it has been described not “a strongly activated” (lane 422) but a defective mitophagic process, as the authors reported in the lane 424 (doi:10.3389/fnmol.2022.921908, doi:10.1016/j.arr.2020.101191)

Minor points:

-       Paragraph 2.1, the authors may mention the recent evidence that the MPTP is associated with, or is an integral part, of a multi-protein complex formed by ATP synthase.

-       Lane 58, there is a comma instead of a dot.

-       Before the citation' bracket, check that a space is present (line 50, 51, 52, 57, 66, etc.) but not a dot (line 44, 63, etc ). The authors need to carefully check the entire manuscript.

-       Line 66, the dot is red

The syntax, the grammar and the English use will need attention. I suggest that a native speaker with editorial input may be helpful.

Particularly:

-       Lanes 440-443 “MtDNA double-strand breaks, as well as progressive loss of intact mtDNA, reduced mtDNA transcription and decreased expression of respiratory chain subunits encoded by the mitochondrial genome were reported in AD cells, which were also observed in ferroptosis mitochondrial.” Grammatically it is not correct (if you use “which”, it refers to the word just before it; in this case AD cells.

It may be more clear: ….decreased expression of respiratory chain subunits encoded by the mitochondrial genome were reported in AD cells, all changes that were also observed in mitochondrial ferroptosis.”

-       Similarly, in the lanes 144-145, 191-194, 249-251, 253-256, 279-280, 290-293, 328-329, 401-403, 436-438, 452-453 and 500, check the use of “which”. For example, in the lane 144-145, it may be more correct: Oxidation of proteins by ROS can result in crosslink and aggregation between proteins, causing their functional alterations, including enzyme…”

-       Lane 448 “ Excessive ROS leads to mitophagy in PD patients, a negative feedback mechanism that plays a crucial role in eliminating ROS by inhibiting oxidative stress. Hence, mitophagy is regarded to be neuroprotective by scavenging ROS generation (Amro et al., 2018; Orellana-Urzua et al., 2020; Wible et al., 2018).” The meaning of this sentence it’s not clear. Please, rephrase it and check that the bibliography cited is correct.

-       Lane 570-573 “Peroxisome proliferator-activated receptor

gamma (PPARγ), a major regulator of lipid metabolism (W. Cai et al., 2018), can be activated by oxidized lipids relevant to the initiation of ferroptosis (Itoh et al., 2008), which reciprocally regulated with Nrf2, participating in the regulation of lipids”. The meaning of this sentence it’s not clear. Please, rephrase it.

Author Response

Reviewer 2

Response: Thanks for your comments on manuscript. We have read the manuscript carefully and revised the manuscript. More revised details can be found in the below.

1、We have numbered the references in the manuscript according to their appearance in the main text and highlighted them in red.

2、We have added the publication year of the article to the bibliography.

3、The references cited in lines 421-424 have been corrected and highlighted in red.

4、We have mentioned that “recent evidence suggests that the MPTP is associated with a multi-protein complex formed by ATP synthase”.

5、In line 58, a dot has been used instead of a comma.

6、We have verified throughout the manuscript that there are spaces before citation parentheses, especially in lines 44, 50, 51, 52, 57, 63, 66, etc.

7、The color in line 66 has been corrected.

8、In line 455-456.The revised sentence, now in red in the manuscript, is: “MtDNA double-strand breaks, as well as progressive loss of intact mtDNA, reduced mtDNA transcription and decreased expression of respiratory chain subunits encoded by the mitochondrial genome were reported in AD cells, all changes that were also observed in mitochondrial ferroptosis.”

9、In line 149-151. The revised sentence, now in red in the manuscript, is: “Oxidation of proteins by ROS can result in crosslink and aggregation between proteins, causing their functional alterations, including enzyme inactivation or modification of ion channel activity.”

10、In line 196-199. The revised sentence, now in red in the manuscript, is: “PGC-1α, the primary regulator of mitochondrial biogenesis, is activated during global cerebral ischemia. It stimulates the expression and activation of several nuclear proteins, including UCP2 and SOD2, leading to increased mitochondrial biogenesis and conferring neuroprotective effects.”

11、In line 258-260. The revised sentence, now in red in the manuscript, is: “AA-PE and AdA-PE were the favored substrates for oxidation, undergoing transformation by 15-LOX into ferroptosis signals, such as PE-AdA-OH and PE-AA-OH, thus actively participating in the execution of ferroptosis.”

12、In line 262-265. The revised sentence, now in red in the manuscript, is: “Arachidonate 15-lipoxygenase (ALOX15) is an enzyme responsible for oxygenating PUFAs and bio-membranes. It exhibits selective and specific activity in inducing the oxidation of AA/AdA-PE, leading to the enzymatic production of AA/AdA-PE-OOHs.”

13、In line 288-289. The revised sentence, now in red in the manuscript, is: “Furthermore, in both cerebral patients and MCAO animal models, there is a decrease in the levels of GSH, accompanied by an elevation in lipid peroxidation.”

14、In line 297-301. The revised sentence, now in red in the manuscript, is:“GPX4, a type of lipid enzyme, converts lipid hydroperoxides into non-toxic lipid alcohols, thereby preventing the accumulation of harmful lipid oxidation products. Deletion or inactivation of GPX4 can lead to lethal ferroptosis and neurological dysfunction, often manifesting as progressive cognitive impairment and impaired behavior in the context of cerebral ischemia.”

15、In line 334-338. The revised sentence, now in red in the manuscript, is: “Furthermore, the acidic environment of cerebral ischemia triggers the dissociation of iron from TF, leading to elevated extracellular iron levels and its transfer to neurons, thereby increasing intracellular iron uptake. Ferritin, TFR1, and DMT1 have all been shown to increase during cerebral ischemia, potentially contributing to iron uptake.”

16、In line 345-349. The revised sentence, now in red in the manuscript, is: “This process significantly elevates ROS production, leading to direct damage to proteins, amino acids, nucleic acids, and membrane lipids, ultimately mediating ferroptosis. Furthermore, lipid peroxidation originates from three pathways: lipid autoxidation catalyzed by iron, the esterification and oxygenation of PUFAs, and the generation of lipid ROS associated with the Fenton reaction.”

17、In line 364-366. The revised sentence, now in red in the manuscript, is: “Based on existing animal studies, mitochondrial dysfunction has been reported in ischemic stroke, and this dysfunction has been correlated with oxygen and glucose deprivation (OGD).”

18、In line409-412. The revised sentence, now in red in the manuscript, is: “In fact, it is widely acknowledged that mitochondria play a role in AD. Mitochondrial ferritin (FtMt) regulates iron metabolism by controlling the transfer of iron from mitochondria to the cytoplasm, thereby safeguarding mitochondria against oxidative damage induced by excessive iron.”

19、In line 448-451. The revised sentence, now in red in the manuscript, is: “It is worth noting that lipid peroxidation and mitochondrial impairments associated with ferroptosis are observed in this model. These effects can be partially mitigated by ferroptosis inhibitors, providing confirmation of the involvement of ferroptosis.”

20、In line 461-467. The revised sentence, now in red in the manuscript, is: “However, dysfunction in mitophagy has been observed in several PD models, leading to an inability to effectively scavenge ROS.”

21、In line 512-514. The revised sentence, now in red in the manuscript, is: “Patients with epilepsy exhibit lower levels of GPX4 and GSH compared to normal controls, and these alterations have been associated with ferroptosis.”

22、In line 461-467. The revised sentence, now in red in the manuscript, is: “Excessive ROS levels in PD patients trigger mitophagy, which is a negative feedback mechanism crucial for reducing oxidative stress by inhibiting ROS production. Consequently, mitophagy is considered neuroprotective as it helps eliminate ROS generation.”

23、In line 583-586. The revised sentence, now in red in the manuscript, is: “Peroxisome proliferator-activated receptor gamma (PPARγ), a key regulator of lipid metabolism, can be activated by oxidized lipids that are relevant to the initiation of ferroptosis. This activation is reciprocally regulated with Nrf2 and plays a role in lipid regulation.”

Round 2

Reviewer 1 Report

I appreciate the authors have addressed all of my comments and concerns in the revised version. I recommend accepting the paper for publication in its present form.

Reviewer 2 Report

Authors followed most of my suggestions making changes and formatting the manuscript.

The manuscript is significantly improved and can now be published in the Brain Sciences.